# The Effect of Surface Oxygen Coverage on the Oxygen Evolution Reaction over a CoFeNiCr High-Entropy Alloy

**DOI:** 10.3390/nano14121058

**Published:** 2024-06-19

**Authors:** Geng Yuan, Luis Ruiz Pestana

**Affiliations:** 1Department of Chemical, Environmental and Materials Engineering, University of Miami, Coral Gables, FL 33146, USA; gxy102@miami.edu; 2Department of Civil and Architectural Engineering, University of Miami, Coral Gables, FL 33146, USA

**Keywords:** oxygen evolution reaction, electrocatalysts, high-entropy alloy, surface oxygen coverage, density functional theory

## Abstract

Developing cost-effective and highly active electrocatalysts for the oxygen evolution reaction (OER) is crucial for advancing sustainable energy applications. High-entropy alloys (HEAs) made from earth-abundant transition metals, thanks to their remarkable stability and electrocatalytic performance, provide a promising alternative to expensive electrocatalysts typically derived from noble metals. While pristine HEA surfaces have been theoretically investigated, and the effect of oxygen coverage on conventional metal electrocatalysts has been examined, the impact of surface oxygen coverage on the electrocatalytic performance of HEAs remains poorly understood. To bridge this gap, we employ density functional theory (DFT) calculations to reconstruct the free energy diagram of OER intermediates on CoFeNiCr HEA surfaces with varying oxygen coverages, evaluating their impact on the rate-limiting step and theoretical overpotential. Our findings reveal that increased oxygen coverage weakens the adsorption of HO* and O*, but not HOO*. As a result, the theoretical overpotential for the OER decreases with higher oxygen coverage, and the rate-limiting step shifts from the third oxidation step (HOO* formation) at low coverage to the first oxidation step (HO* formation) at higher coverage.

## 1. Introduction

The oxygen evolution reaction (OER), which oxidizes H_2_O to O_2_, is crucial for various sustainable energy technologies. For instance, electrochemical water splitting is a promising and environmentally friendly method for hydrogen production [1,2,3], whose efficiency is notably hindered by the high overpotential of the OER—its anodic half-reaction [4,5]. While noble metal-based oxides like IrO_2_ and RuO_2_ are currently the most practical OER electrocatalysts [6,7,8], with experimental overpotentials typically within the range of 200–300 mV at a current density of 10 mA cm^−2^ [9,10], their broader utilization in water splitting is limited by the high cost and scarcity of noble metals. In the pursuit of cost-effective OER electrocatalysts, high-entropy alloys (HEAs) have gained increasing attention due to their excellent OER activity and electrochemical stability [11,12,13]. HEAs are materials composed of four or more metals in roughly equal proportions, capable of forming single-phase solid solutions stabilized by the entropy of mixing [14,15,16]. Unlike conventional electrocatalyst materials, such as noble metals or their oxides, which have limited tunability in their catalytic activity, HEAs stand out due to their vast design space, the immense diversity of local binding environments that could act as active sites, and exceptional stability [14,17,18]. 

Considering the prevalence of earth-abundant transition metals such as Co, Fe, and Ni in OER electrocatalysts [19,20,21,22,23,24,25,26,27], HEAs developed for this purpose commonly incorporate two or three of these metals in their chemical composition, substituting noble metals [11,12]. For instance, Jin et al. investigated a nanoporous AlNiCoIrMo HEA [28], reporting both its excellent electrochemical stability and low overpotential of 233 mV at 10 mA cm^−2^. Studies on noble metal-free HEAs have also revealed promising results. Qiu et al. explored the effect of alloying various non-noble metals with CoFeNi on the OER performance, highlighting the AlNiCoFeMo HEA as a particularly noteworthy composition due to its significantly enhanced electrochemical durability and the lowest measured overpotential in that study of about 240 mV at 10 mA cm^−2^ [29]. Waag et al. demonstrated that the CoCrFeMnNi HEA exhibits comparable performance to commercial IrO_2_, with an average mass-specific activity of 0.104 A/mg [30]. Huang et al. found that lattice distortion in MnFeCoNiCu enhances its catalytic activity, leading to an overpotential of 263 mV at 10 mA cm^−2^ in the OER [31]. In addition to quinary HEAs, Dai et al. reported the promising OER performance of a quaternary MnFeCoNi HEA-based electrocatalyst, with an overpotential of 302 mV at the current density of 10 mA cm^−2^, comparable to that of RuO_2_ [32]. These experimental findings demonstrate the promise of earth-abundant transition metal HEAs for the OER. Besides HEAs, important theoretical and experimental work has also been conducted on evaluating the electrocatalytic properties of perovskite oxides, relatively new materials known for their versatile structure and tunable properties. This includes identifying universal activity descriptors [33], or exploiting lattice strain to tailor their adsorption characteristics and enhance their catalytic properties [34].

Previous theoretical studies using density functional theory (DFT) have shed light on the OER mechanisms and catalytic performance of HEAs by analyzing the free energy of OER intermediates on pristine HEA surfaces [35,36,37,38,39]. However, these studies have focused on pristine surfaces and overlooked the effect of surface oxygen coverage expected during the OER [6], which remains unknown. Considering the effect of oxygen coverage is of great importance as it can significantly influence the stability of OER intermediates and, consequently, affect the electrocatalytic performance of the HEA. The formation of an oxygen layer on the surface of the HEA during operation is likely due to the oxophilicity of pristine HEA surfaces made of transition metals like Co, Fe, Ni, and Cr [40].

Here, we use DFT calculations to study the effect of surface oxygen coverage on the adsorption energies on a CoFeNiCr HEA of OER intermediates (hydroxyl (HO*), oxygen (O*), and hydroperoxyl (HOO*) radicals), and we construct a free energy diagram based on the computational hydrogen electrode (CHE) model [41]. Through thermodynamic analysis, we gain insight into the impact of surface oxygen coverage on the theoretical OER overpotential and the associated rate-limiting step. The composition CoFeNiCr was selected because its constituent metals are among the most frequently utilized in HEAs [42], it has desirable mechanical properties [43], and it has shown promising results for hydrogen evolution reactions (HER) [44], making it a potential candidate for exploring dual-functionality in electrocatalytic applications. 

## 2. Materials and Methods

DFT calculations were conducted using the Quickstep module of the CP2K 9.1 simulation package [45], which employs a dual basis of atom-centered Gaussian orbitals and Plane Waves known as the mixed Gaussian and plane wave (GPW) approach. The exchange-correlation functional revPBE-D3 was used, which integrates the generalized-gradient approximation (GGA) functional revPBE (revised Perdew–Burke–Ernzerhof) [46] with Grimme’s DFT-D3 dispersion corrections with zero-damping [47]. The core electrons of all the elements were represented by Goedecker–Teter–Hutter (GTH) pseudopotentials (PPs) [48,49], which have been shown to perform well across different systems and chemical conditions while also being computationally efficient for the GPW approach implemented in CP2K. The molecularly optimized (MOLOPT) Gaussian basis sets (BBs), which were specifically developed for use with GTH-PPs, were employed [50]. Specifically, the triple-zeta valence molecularly optimized basis set (TZV2P-MOLOPT-GTH) was used for H and O, which, as our previous work shows, offers a good balance between accuracy and computational cost [40]. This basis set is also the preferred choice for simulating ab initio liquid water [51,52]. The double-zeta valence short-range molecularly optimized basis set (DZVP-MOLOPT-SR-GTH) was used for Co, Fe, Ni, and Cr, as it is the largest available for those elements. The PPs and BSs used in this study were also previously employed to simulate the oxidation of copper [53].

To facilitate the convergence of the calculations, Fermi–Dirac smearing with an electronic temperature of 500 K was applied, along with the use of Broyden mixing [54]. The Brillouin zone was sampled at the Gamma point and periodic boundary conditions were applied in all three dimensions. The plane-wave energy cutoff for all calculations was set at 400 Ry. All DFT calculations involving oxygen intermediates were performed with spin polarization, with the only unpaired electrons being on the adsorbate. The HEAs were treated as non-magnetic. Previous research has reported a Curie temperature of approximately 130 K for the CoFeNiCr HEA [55,56]. Since the electrocatalytic process typically occurs at room temperature, we hypothesize that the magnetic properties of the HEA will have a modest impact under operating conditions. Furthermore, predicting the magnetic ground state of HEAs presents significant challenges due to their local compositional heterogeneity, which is beyond the scope of this paper.

The simulations featured slabs of a face-centered cubic (FCC) CoFeNiCr HEA with an FCC (111) surface and varying surface oxygen coverages. The HEA slabs were 4 × 4 atoms wide and 6 atoms thick, totaling 96 metal atoms. We assumed a random model for the HEA, where the atoms in the lattice were randomly assigned to Co, Fe, Ni, or Cr, maintaining the equimolar stoichiometry of the HEA. While this approximation is not entirely accurate, as theoretical studies have shown a tendency for Cr to segregate [57,58], it is a commonly used approximation to model HEAs. We expect Cr segregation to affect minor quantitative details, but not the qualitative trends reported here.

The following steps were carried out to generate each system. First, a bulk 4 × 4 × 6 HEA was optimized allowing adjustments to both the simulation box and the atomic positions using the Broyden–Fletcher–Goldfarb–Shanno (BFGS) algorithm [59] with a convergence criterion of 10^−5^ Hartree/Bohr. Next, a 10 Å vacuum layer was introduced along the z-direction of the simulation box to effectively simulate a slab with an FCC (111) surface. The slab’s geometry was then optimized, using the BFGS algorithm with a convergence criterion set at 10^−3^ Hartree/Bohr, permitting only the top three layers of atoms to relax. After optimizing the slab’s geometry, atomic oxygens were positioned on the FCC hollow sites—widely recognized as preferred binding sites for atomic oxygen on FCC (111) surfaces [60,61,62,63]—approximately 2 Å above the HEA surface. The positions of the surface oxygen atoms were optimized keeping the metal atoms fixed. Any oxygen atom that was not chemisorbed to the surface after the geometry optimization was removed from the system.

The surface oxygen coverage, $\theta_{O}$, is defined as the ratio of chemisorbed oxygen atoms to the total number of FCC hollow sites on the HEA surface. Thus, it is expressed as a fraction of a monolayer (ML). Considering the slightly fluctuating nature of the number of chemisorbed oxygen atoms on the HEA surfaces after optimization, we categorized $\theta_{O}$ into three levels, ranging from low to high coverage, with $\theta_{O}$ values falling within the ranges [0.25, 0.5), [0.5, 0.75), and [0.75, 1), respectively. Figure 1a displays top-view representations of optimized systems corresponding to different levels of surface oxygen coverage.

Once the oxygen-covered HEA slabs were generated, we added the respective OER intermediates as adsorbates at random positions on the surface and about 3 Å above it. We optimized the geometry of the combined systems keeping the HEA and chemisorbed oxygen atoms fixed. Finally, the adsorption energy (∆E) of each intermediate was calculated using the following equations [9,64]:(1)$$∆E_{HO*}=E_{HEA+HO*}-E_{HEA}+\left(\frac{1}{2}E_{H_{2}}-E_{H_{2}O}\right)$$
(2)$$∆E_{O*}=E_{HEA+O*}-E_{HEA}+\left(E_{H_{2}}-E_{H_{2}O}\right)$$
(3)$$∆E_{HOO*}=E_{HEA+HOO*}-E_{HEA}+\left(\frac{3}{2}E_{H_{2}}-2E_{H_{2}O}\right)$$
where $E_{HEA+HO*}$, $E_{HEA+O*}$, and $E_{HEA+HOO*}$ represent the energies of the combined systems after geometry optimization, $E_{HEA}$ is the energy of the pristine or oxygen-covered slab, and $E_{H_{2}}$ and $E_{H_{2}O}$ are the energies of optimized H_2_ and H_2_O molecules in the gas phase. All the energies were corrected for basis set superposition errors (BSSEs). The adsorption free energy (∆G) of each intermediate was obtained by correcting the adsorption energy (∆E) for the zero-point energy and entropy using the following equation [41]:(4)∆G=∆E+∆ZPE−T∆S
where ∆G represents the change in reaction Gibbs free energy of each intermediate; ∆ZPE and T∆S denote the difference in the zero-point energy and entropy, respectively. The zero-point energies and entropic corrections are taken from reference [41].

Appendix A provides a detailed breakdown of the number of adsorption energy calculations for each intermediate at each $\theta_{O}$, including the number of different HEA configurations, oxygen-covered configurations, and adsorbate configurations used in this study. In total, we carried out over 1200 DFT adsorption energy calculations.

## 3. Results

### 3.1. Non-Uniform Metal Oxidation 

Our simulations revealed that oxygen atoms do not chemisorb equally to the different metals. As illustrated in Figure 1b, which shows the number of occurrences where the various constituent metals were found to be nearest neighbors to chemisorbed oxygen atoms, Cr exhibits the highest occurrence, followed by Fe and Co, while Ni shows the lowest occurrence. This trend reflects the high oxidation tendency of Cr and the low oxidation tendency of Ni, aligning with recent findings on the metal oxidation propensity in the CoFeNiCr HEA [44,65]. This non-uniform oxidation tendency is also reflected on the adsorption behavior of each constituent metal toward oxygen or oxygen-containing adsorbates. Specifically, Ni exhibits the weakest adsorption among all four elements [40,41], making surface Ni atoms the least likely to be the nearest neighbors of chemisorbed oxygens.

### 3.2. Surface Oxygen Coverage Impact on OER Intermediate Adsorption

Due to the wide diversity of binding sites on HEA surfaces, the adsorption energy of OER intermediates cannot be represented by a single value, as is typical for conventional electrocatalysts with one or a few active sites. Instead, it must be represented by a distribution of adsorption energies. Figure 2a–c show the distributions of adsorption energies for HO*, O*, and HOO*, respectively, at each level of surface oxygen coverage (top to bottom). The distributions display two distinct modes. Mode I, indicated by orange arrows in Figure 2a, corresponds to binding configurations where the adsorbate occupies a pristine site on the HEA surface, while Mode II, indicated by green arrows in Figure 2a, involves binding configurations where the adsorbate is on or near a site already occupied by a chemisorbed oxygen atom. Figure 2d provides snapshots from the simulations, illustrating both modes.

The most straightforward effect of surface oxygen coverage on the adsorption energies is its impact on the population of both modes. On pristine surfaces ($\theta_{O}=0$), Mode II obviously does not exist. As $\theta_{O}$ increases, the number of available pristine binding sites on the surface decreases, leading to fewer Mode I and more Mode II occurrences. This unsurprising effect on the distributions is readily apparent in Figure 2a–c, progressing from the bottom to the top panels.

More interestingly, $\theta_{O}$ has a non-trivial effect, different for each adsorbate, on the average adsorption energy associated with each mode (Figure 2e–g). For both HO* (Figure 2e) and O* (Figure 2f), as $\theta_{O}$ increases, the average adsorption energy associated with Mode I shifts significantly towards weaker adsorption, while that associated with Mode II remains largely unaffected by surface oxygen coverage. The substantial shift in the average adsorption energy associated with Mode I for both HO* and O* suggests that adjacent chemisorbed oxygen atoms weaken their adsorption at pristine sites. This phenomenon has been previously reported for single metals [66,67]. In contrast, the average adsorption energy associated with Mode II for all intermediates remains mostly constant irrespective of $\theta_{O}$, indicating that adsorption at oxidized sites is minimally affected by other nearby chemisorbed oxygens. For HOO*, the average adsorption energy associated with both modes remains consistent across different $\theta_{O}$ ranges (Figure 2g). However, unlike for HO* and O*, increasing $\theta_{O}$ significantly impacts the frequency of spontaneous HOO* dissociation. Our simulations reveal that on pristine HEA surfaces, the O-O bond in HOO* dissociates in 95% of the cases during geometry optimization. This frequency drops to approximately 50% when $\theta_{O}$ increases to the range [0.25, 0.5) and further declines to around 14% when $\theta_{O}$ is higher than 0.5 ML. This phenomenon underscores the strong tendency of HEA surfaces to oxidize [68].

In an effort to gain mechanistic understanding of the adsorption of HO*, we examined how the tilt angle, α, of HO* with respect to the surface of the HEA varies depending on the oxygen coverage and whether HO* is adsorbed to a pristine or oxidized site (Figure 3a). According to our definition, when α=0°, HO* is vertical on the surface with the hydrogen pointing away from it. For pristine sites (i.e., Mode I), HO* typically adopts a mostly vertical configuration at hollow sites (α≈0° to 30°) and a more tilted configuration at bridge sites (α≈30° to 60°), as evidenced by the bimodal distributions for pristine sites shown in blue in Figure 3a. The average tilt angle at pristine hollow sites increases with oxygen coverage, from approximately 5° on pristine surfaces to about 15° for $\theta_{O}\in [0.25,\;0.5)$. For Mode II adsorption, HO* configurations range from parallel to the surface (α=90°) to having the hydrogen in HO* pointing towards the HEA surface (α>90°). Interestingly, Figure 3b reveals a correlation between adsorption energy and tilt angle, with smaller tilt angles resulting in stronger adsorption energies.

A recent study by Cao and Nørskov [69] demonstrated that spin polarization results in consistently weaker adsorption energies for a wide range of adsorbates on single metal surfaces such as Co, Fe, and Ni. We hypothesize that similar trends would be observed for the CoFeNiCr HEA in this study if magnetic effects were considered, particularly for the adsorption to pristine sites. Given that this weakening effect was uniform across all adsorbates studied, it is reasonable to believe that the qualitative trends we observe here would remain unchanged.

### 3.3. Free Energy Diagram for the OER

To assess how changes in the adsorption energy of each intermediate as $\theta_{O}$ varies might influence the electrocatalytic performance of HEAs, we constructed and analyzed the free energy diagrams for the OER, assuming it proceeds through the four-electron pathway mechanism:(5)$$2H_{2}O\to HO*+H_{2}O+H^{+}+e^{-}$$
(6)$$HO*+H_{2}O+H^{+}+e^{-}\to O*+H_{2}O+{2H}^{+}+{2e}^{-}$$
(7)$$O*+H_{2}O+{2H}^{+}+{2e}^{-}\to HOO*+{3H}^{+}+{3e}^{-}$$
(8)$$HOO*+{3H}^{+}+{3e}^{-}\to O_{2}+{4H}^{+}+{4e}^{-}$$

The free energy difference (∆G) of each oxidation step i (i = 1, 2, 3, or 4) at pH = 0 and zero potential (U = 0 V) is calculated as follows:(9)$$∆G_{1}=∆G_{HO*}$$
(10)$$∆G_{2}=∆G_{O*}-∆G_{HO*}$$
(11)$$∆G_{3}=∆G_{HOO*}-∆G_{O*}$$
(12)$$∆G_{4}=∆G_{O_{2}}-∆G_{HOO*}$$

Figure 4 contains the free energy diagrams of the OER on HEAs for the three surface oxygen coverage ranges studied here. The analysis excludes pristine HEA surfaces as HOO* was found to dissociate in most cases on those surfaces. In Figure 4, we employ box plots to visually capture the spread of adsorption free energies for each intermediate, a representation unnecessary for conventional catalysts. 

By analyzing the median free energy of each reaction step as a function of $\theta_{O}$, we observe a significant increase in $∆G_{1}$, the free energy difference for the initial oxidation step from H_2_O to HO*, as $\theta_{O}$ increases. This increase in $∆G_{1}$ is due to the overall weakening of HO* adsorption, which results from both the diminished number of available pristine binding sites on the surface and the reduced adsorption strength at the remaining pristine sites. The free energy difference $∆G_{2}$, the step from HO* to O*, also rises as $\theta_{O}$ increases, albeit not as much as the increase observed for $∆G_{1}$. The simultaneous weakening of HO* and O* adsorption on surfaces with higher oxygen coverages accounts for the observed trend. Since the adsorption of HOO* remains more or less the same across different oxygen coverages, the overall reduction in O* adsorption strength results in a significant decrease in the free energy difference $∆G_{3}$, the third oxidation step from O* to HOO*, on surfaces with higher oxygen coverages. 

### 3.4. Theoretical Overpotential for OER

The theoretical overpotential ($\eta^{OER}$) is calculated as the difference between the highest free energy change among all four oxidation steps and the theoretically minimum potential difference required for the reaction to proceed (1.23 V):(13)$$\eta^{OER}=\frac{\max\left[∆G_{1},∆G_{2},∆G_{3},∆G_{4}\right]}{e}-1.23V$$In contrast to the previous section, which focused on the median free energy values of the distributions, this section calculates $∆G_{1},∆G_{2},∆G_{3},∆G_{4}$, and therefore $\eta^{OER}$, at the level of individual binding sites on each HEA. Specifically, we only include the subset of cases where all three intermediates, after optimization, are adsorbed to the same binding site on the same oxygen-covered HEA surface. We consider two intermediates to be adsorbed to the same site if the relative distance between the centers of the oxygen atoms is less than 1.5 Å (which is approximately the distance between an adsorbed oxygen atom at a pristine site and the nearest neighbor metallic elements). Figure 5a shows snapshots of an example where all the intermediates were found to be adsorbed to the same binding site on the HEA surface. The number of such sites that we were able to sample is quantified in Table 1.

The distribution of $\eta^{OER}$ for each $\theta_{O}$ range is shown in Figure 5b. We find that $∆G_{4}$ is never the rate-limiting step for any of the binding sites studied here, and $∆G_{2}$ serves as the rate-limiting step in less than around 5% of the cases. The data in Table 1 suggest a strong correlation between the type of binding site (i.e., pristine site or Mode I versus oxidized site or Mode II) and the rate-limiting step of the reaction. Pristine sites predominantly result in $∆G_{3}$ being the highest free energy difference, which corresponds to the step from O* to HOO*. Conversely, oxidized sites typically lead to $∆G_{1}$ being the highest free energy difference, corresponding to the initial oxidation step from H_2_O to HO*. As the number of oxidized sites increases at the expense of pristine sites with increasing $\theta_{O}$, the rate-limiting step shifts from $∆G_{3}$ to $∆G_{1}$.

In Figure 5b, it is clear that the $\eta^{OER}$ associated with $∆G_{3}$ (yellow), typically linked to pristine sites, displays a broader distribution than that associated with $∆G_{1}$ (blue). This is because the composition of the binding site has a greater impact on pristine sites than on oxidized sites. The $\eta^{OER}$ associated with $∆G_{3}$ (i.e., pristine binding sites) can range from as low as 0.96 V observed at $\theta_{O}\in [0.5,\;0.75)$ to as high as 4.97 V observed at $\theta_{O}\in [0.25,\;0.5)$. We observe a shift in the rate-limiting step as $\theta_{O}$ increases (Figure 5c). The proportion of rate-limiting steps attributed to $∆G_{3}$ decreases from about 63% at the lowest oxygen coverage to about 10% for $\theta_{O}\in [0.75,\;1)$, while that attributed to $∆G_{1}$ increases from about 32% to over 90% across the same ranges of oxygen coverage.

Our mechanistic understanding for the shift in the rate-limiting step is as follows: Our simulations show that the adsorption of HOO* remains largely unaffected by $\theta_{O}$. In contrast, the adsorption of HO* and O*, which is initially strong at low $\theta_{O}$, weakens as $\theta_{O}$ increases. As mentioned earlier, this weakening is due to the decreasing number of available pristine sites and the reduced adsorption energy of the remaining pristine sites. Consequently, at low $\theta_{O}$, the rate-limiting step is the third oxidation step (from O* to HOO*), primarily due to the weak adsorption of HOO*. As both HO* and O* adsorption weaken with increasing $\theta_{O}$ (while HOO* adsorption remains unchanged), there comes a point where the adsorption of HO* becomes so weak that the first oxidation step (from H2O to HO*) becomes the rate-limiting step.

When examining the average overall overpotential, $\bar{\eta_{OER}}$, at each oxygen coverage range (Figure 5d), we observe a general decrease as $\theta_{O}$ increases. This suggests that the catalytic performance towards the OER improves with higher oxygen coverage. This improvement is primarily driven by the decreasing average overpotential associated with $∆G_{3}$, as indicated in the inset of Figure 5d. In contrast, the average overpotential associated with $∆G_{1}$ remains consistent, regardless of oxygen coverage.

## 4. Conclusions

In this study, we used DFT simulations to investigate the effect of surface oxygen coverage ($\theta_{O}$) on a CoFeNiCr HEA in the oxygen evolution reaction (OER). We found that as surface oxygen coverage increases, the adsorption of intermediates HO* and O* weakens notably. This weakening is attributed to the rise in the number of oxidized binding sites, which exhibit weaker adsorption than pristine sites, and the reduced adsorption on the remaining pristine sites due to the influence of neighboring chemisorbed oxygens. Interestingly, the adsorption energy of HOO* is mostly independent of the level of oxygen coverage. This non-uniform effect on the adsorption energies of the different intermediates causes a transition in the rate-limiting step from the third oxidation step (formation of HOO*) to the first oxidation step (formation of HO*) as surface oxygen coverage rises. Furthermore, the average theoretical overpotential decreases as $\theta_{O}$ increases.

Our work marks a significant step towards understanding the electrocatalytic potential of HEAs. Beyond their use in the OER, our simulations reveal a wide range of adsorption energies and overpotentials associated with the diversity of binding sites, highlighting their potential as multifunctional electrocatalysts.

## Figures and Tables

**Figure 1 nanomaterials-14-01058-f001:**
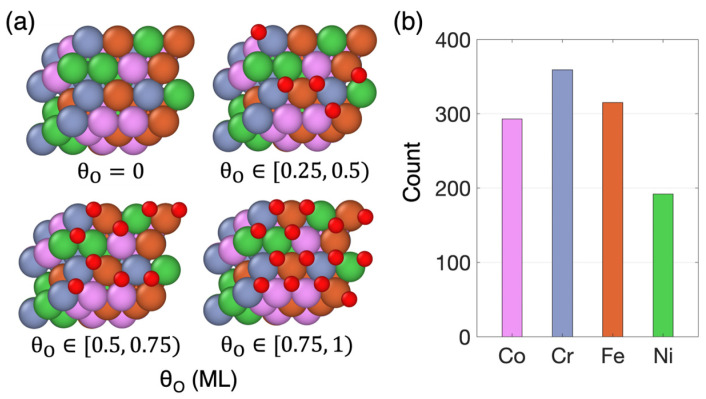
(**a**) Top view of optimized oxygen-covered HEA configurations at different $\theta_{O}$ ranges, including the pristine surface configuration ($\theta_{O}=0)$. The red atoms represent chemisorbed oxygen atoms, while pink, blue, orange, and green correspond to Co, Cr, Fe, and Ni atoms, respectively. (**b**) The number of occurrences where the various constituent metals were found to be nearest neighbors to chemisorbed oxygen atoms.

**Figure 2 nanomaterials-14-01058-f002:**
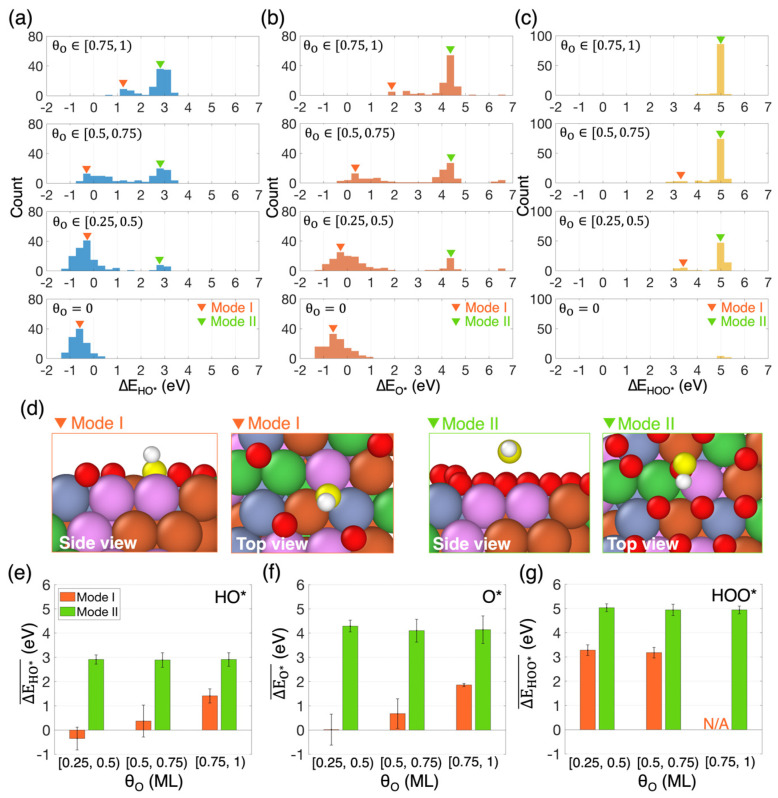
(**a**–**c**) Adsorption energy distributions for HO*, O*, and HOO*, respectively, on HEA surfaces with different $\theta_{O}$, including pristine surfaces (bottom, $\theta_{O}=0$). The orange and green arrows point to the peaks corresponding to Mode I and Mode II, respectively. The data visible in some of the distributions in panel (**b**), around 6–7 eV, correspond to desorbed O*. (**d**) Snapshots illustrating side and top views of example binding configurations of HO* corresponding to Mode I and Mode II of the distributions. The oxygen atom of HO* is colored in yellow. (**e**–**g**) Average adsorption energy of HO*, O*, and HOO*, respectively, for both modes as a function of the oxygen coverage. The error bars represent standard deviation.

**Figure 3 nanomaterials-14-01058-f003:**
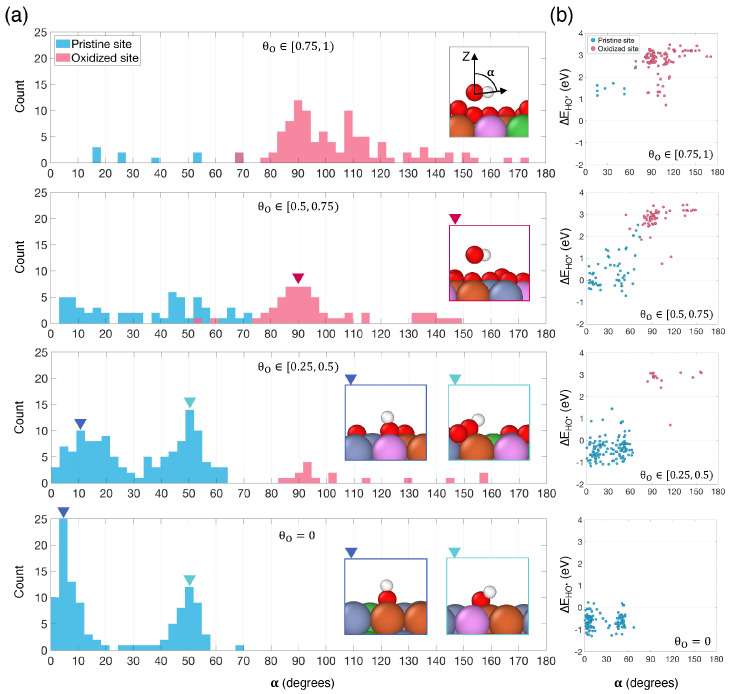
(**a**) Distribution of the tilt angle, α, of HO* with respect to the HEA surface for different oxygen coverages and for HO* adsorbed to pristine (blue) or oxidized (red) sites. The inset in the top plot illustrates the definition of α. Insets in the other plots show examples of the configuration of HO* corresponding to some notable peaks in the distributions, indicated by arrows. (**b**) Correlation between $\Delta E_{HO*}$ and α for different oxygen coverages, with HO* adsorbed to pristine (blue) or oxidized (red) sites.

**Figure 4 nanomaterials-14-01058-f004:**
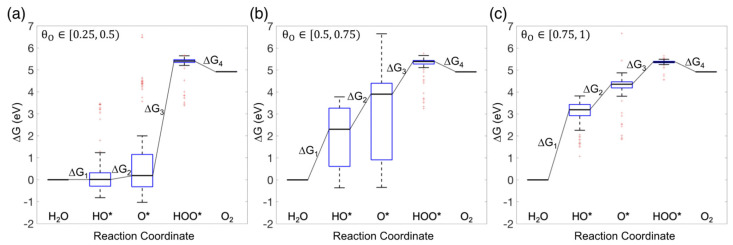
Free energy diagrams for oxygen evolution reaction (OER) on oxygen-covered HEA surfaces at zero potential. (**a**) $\theta_{O}\in [0.25,\;0.5)$, (**b**) $\theta_{O}\in [0.5,\;0.75)$, and (**c**) $\theta_{O}\in [0.75,\;1)$. Each box plot represents the distribution of adsorption free energies across all binding site compositions for each intermediate. The solid line within each box plot represents the median adsorption free energy.

**Figure 5 nanomaterials-14-01058-f005:**
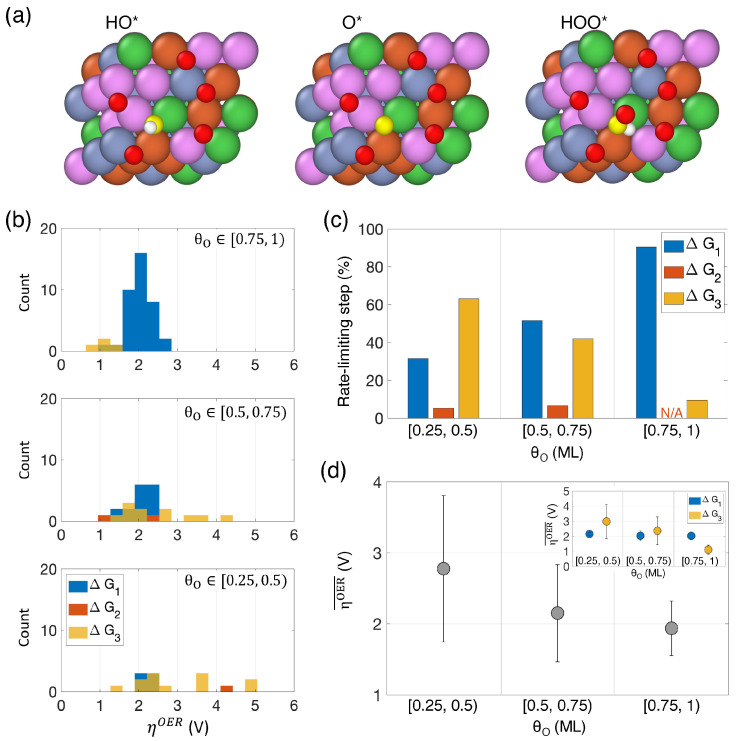
(**a**) Snapshots of an example where all the intermediates were found to be adsorbed to the same binding site on the HEA surface. The central oxygen atoms of HO*, O*, and HOO* are highlighted in yellow. (**b**) Distributions of theoretical overpotentials ($\eta^{OER}$) for each $\theta_{O}$ range. The $\eta^{OER}$ values corresponding to different rate-limiting steps in the OER are depicted in blue, orange, and yellow, representing $∆G_{1}$, $∆G_{2}$, and $∆G_{3}$, respectively. (**c**) Percentage of times that $∆G_{1}$, $∆G_{2}$, or $∆G_{3}$ were the rate-limiting steps as a function of $\theta_{O}$. (**d**) Average theoretical overpotential ($\bar{\eta_{OER}}$) accounting for contributions from all rate-limiting steps. The error bars indicate the standard deviation of the total distributions shown in Panel (**b**). The inset shows the $\bar{\eta_{OER}}$ as a function of $\theta_{O}$ corresponding to the individual rate-limiting steps $∆G_{1}$ and $∆G_{3}$.

**Table 1 nanomaterials-14-01058-t001:** Number of individual binding sites studied for calculating the theoretical overpotential (η^OER^) at different coverage ranges. The table provides detailed information on the number of individual binding sites corresponding to either pristine binding site (Mode I) or oxidized binding site (Mode II) with oxidation step i (i = 1, 2, or 3) as the rate-limiting step.

$\theta_{O}$ (ML)	Number of Individual Sites	Number of Pristine Sites			Number of Oxidized Sites		
		Rate-Limiting Step			Rate-Limiting Step		
		$∆G_{1}$	$∆G_{2}$	$∆G_{3}$	$∆G_{1}$	$∆G_{2}$	$∆G_{3}$
[0.25, 0.5)	19	N/A	1	12	6	N/A	N/A
[0.5, 0.75)	31	2	1	13	14	1	N/A
[0.75, 1)	42	N/A	N/A	1	38	N/A	3

## Data Availability

The original contributions presented in the study are included in the article/Appendix A, further inquiries can be directed to the corresponding author.

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
