# Peer review of "The Effect of Surface Oxygen Coverage on the Oxygen Evolution Reaction over a CoFeNiCr High-Entropy Alloy"

_nanomaterials, 2024, doi:10.3390/nano14121058_

Round 1

Reviewer 1 Report

Comments and Suggestions for Authors

The author presents a study on the influence of surface oxygen coverage on the oxygen evolution reaction (OER) using CoFeNiCr high-entropy alloy (HEA). The study employs density functional theory (DFT) calculations to explore the changes in free energy diagrams and the rate-limiting steps under varying oxygen coverages. The findings suggest that increased oxygen coverage leads to a decreased theoretical overpotential for OER, driven by changes in the adsorption energies of key intermediates. However, there are some aspects where revisions are recommended:

1.     The paper utilizes the CP2K simulation package for DFT calculations but lacks a detailed explanation of the choice of pseudopotentials and basis sets, particularly how they impact the accuracy of the results. Could the authors elaborate on their methodological choices, especially in the context of their suitability for HEA systems?

2.     Given that magnetic properties can influence the electronic structure and catalytic performance, could the authors discuss the potential impact of ignoring magnetic interactions in their simulations?

3.     The study presumes a uniform distribution of metals in the HEA surface. However, HEAs are known for their local compositional heterogeneity. How might this heterogeneity affect the adsorption energies and the overall conclusions drawn in the study?

4.     The shift in the rate-limiting step with increased oxygen coverage is a crucial finding. Could the authors provide a more detailed mechanistic explanation of why this shift occurs?

5.     It would be beneficial to include a more quantitative analysis or a comparative study with other OER materials. (Such us: Adv. Mater. 2023, 35, 2305074; J. Energy Chem. 82, 572-580; Appl. Phys. Rev. 9, 011422, which could be mentioned)

Comments on the Quality of English Language

Minor editing of English language required

Reviewer 2 Report

Comments and Suggestions for Authors

The oxygen evolution process (OER) on CoFeNiCr high-entropy alloy (HEA) surfaces is assessed in this work using density functional theory (DFT) calculations. It is discovered that increasing oxygen coverage lowers the theoretical overpotential because of weaker adsorption of HO* and O* intermediates. The rate-limiting phase shows how surface oxygen coverage significantly affects OER performance in HEAs, shifting from HOO* formation at low coverage to HO* formation at greater coverage. The following question should be answered before the acceptance of the article in the Nanomaterials.

1.     The novelty of this study should be inserted in the abstract.

2.     The introduction seems very short and needs to develop with new findings. Also, it should be compared with existing literature is much appreciated.

3.     The authors should explain the impact of the materials selected in this work.

4.     For CoFeNiCr HEA surfaces, what is the effect of increased oxygen coverage on the theoretical overpotential for the OER?

5.     What essential intermediates are taken into consideration in the free energy diagram of the OER on HEA surfaces.

6.     A variable oxygen coverage during the OER on CoFeNiCr HEA does not impact which intermediate?

7.     What is the correlation between the rate-limiting step and the various oxygen coverages on the HEA surface?

8.  For OER applications, why is it important to thoroughly investigate whether oxygen coverage impacts HEAs?

9.     why did the author specifically select these materials for DFT investigation? Numerous materials have been examined for HEA materials. The study's author ought to compare it to respectable literature.

10.  In what way do the results of this study on HEAs differ from those of traditional electrocatalysts made of noble metals?

11.  I suggested bringing all the SI pieces of information to the main articles and  the discussion.

12.  There are some typographical and grammatical errors in the manuscript. Hence, the manuscript should be carefully checked and necessary corrections should be made.

13.  The author has to replace "Fig" in the caption with "Figure' uniformly throughout the manuscript to make the uniformity.

14.  The images are not clear. It seems compressed. Make it clear for the readers.

15.  Update references by adding some recent references supporting your work.

Comments on the Quality of English Language

The English language corrections are mandatory in this journal. 

Round 2

Reviewer 2 Report

Comments and Suggestions for Authors

The manuscript is now ready for publication after the author has addressed all the comments.

Comments on the Quality of English Language

Minor editing of the English language required